# Scenarios for Reducing Greenhouse Gas Emissions from Food Procurement for Public School Kitchens in Copenhagen

Adam Addis Prag, Julie Bangsgaard Abrahams, Filippo Daniele, Maya S. Dodhia, Chujie Feng, Kevin Hahn, Steffen Kristiansen, Anna Maria Leitner, Jordi Pedra Mendez, Marcel Mohr, Sofie Fønsskov Møller, Simon Yde Svensson, Kea-Lena Permin Talbot, Ilie Tomulescu, Barbora Valachova, Fatimah Zahra, Marin Lysák and Christian Bugge Henriksen *

Department of Plant and Environmental Science, University of Copenhagen, 2630 Taastrup, Denmark; ilie@peopleof2050.org (I.T.)
* Correspondence: cbh@plen.ku.dk

**Abstract:** The food system is responsible for a third of global greenhouse gas emissions, with the majority originating from livestock. Reducing our meat consumption is thus an important part of achieving necessary reductions in emissions, and reaching children is especially important to facilitate long-lasting changes in dietary habits now and into the future. This study developed dietary scenarios for three public schools in Copenhagen, which were used as cases to demonstrate reduction in greenhouse gas emissions from public kitchens. The scenarios included (i) replacement of all beef with poultry, (ii) replacement of all meat and fish with legumes, and (iii) alignment of food procurement to the Danish Food Based Dietary Guidelines based on the Planetary Health Diet. The effects on emissions were calculated using three different LCA databases. The results showed reductions ranging from 32 to 64% depending on the scenario, the current meal plan at the case school, and the emission factors used. Not surprisingly, the vegetarian scenario resulted in the highest reductions and replacing beef resulted in the lowest. Adhering to the national guidelines will result in reductions in emissions of 39–48%. Significant variability in the results existed between the three databases, highlighting the importance of basic understanding of LCA for kitchens interested in estimating and reducing their carbon footprint while at the same time providing justification for applying multiple LCA databases for increasing robustness.

**Keywords:** climate impact of food; sustainable food; plant-rich diet; climate-friendly diet; school meals; reducing greenhouse gas emissions

## 1. Introduction

The current global food system is responsible for up to a third of total global greenhouse gas (GHG) emissions [1,2]. The majority of these emissions are due to livestock farming, both as direct emissions, especially from ruminants, and due to soil emissions from large areas that are used for the cultivation of animal feed [3,4]. One estimate suggests that 20% of the total global GHG emissions are directly related to livestock [5], and a recent study found that phasing out animal agriculture would contribute half of the reductions necessary to limit global warming to 2.0 °C by 2100 [6]. Reducing the climate impact of our food systems is thus vital to reach the reduction targets specified in the Paris Agreement, and global transition to a more plant-based diet is an important option for achieving this aim [6–8].

In Denmark, the municipality of Copenhagen is currently pursuing the aim of reducing GHG emissions from its public kitchens by at least 25% per capita by 2025 relative to 2018 levels [9], and the ambition of the target is expected to increase. Most municipalities and regions in Denmark are not yet at a point in their sustainability efforts where they have started considering quantified reductions in food-related emissions and dietary

changes [10]. Quantifying a reduction in emissions from public kitchens requires both attention to the different conditions in the institutions providing meals and an understanding of how the choice of methodology impacts the estimated benefit of dietary interventions. Generally, the choice of LCA approach has the potential to significantly affect the result, especially when assessing production systems with multiple products to which the total environmental impacts of production must be allocated [11–13]. The estimated effect of dietary changes and the relative importance of alterations in food procurement for public kitchens can thus be significantly influenced by the choice of the assessment method.

Additionally, to eventually achieve lasting and society-wide transformation of the average citizen's diet, attention to food habits is required. Dietary habits formed in childhood are believed to last into adulthood, and the formation of these habits and preferences are heavily influenced by the food environment during childhood [14]. It is well known that the socioeconomic status and the food consumed at home affects the future dietary habits of children [15], and providing healthy and sustainable alternatives in schools and other institutions can potentially aid in the development of better habits. Developing the taste for a sustainable diet in childhood can be an important part of gradually achieving more sustainable preferences in the population as a whole. Pertinently placed for this, Copenhagen has 15 so-called "Food Schools". In this subset of the municipality's 70 public schools, lunch is prepared on the school premises instead of being delivered from a central kitchen, and food education and participation in preparation of the food is incorporated into the curriculum to varying extents [16]. The Food Schools are part of a larger trend that has taken place in public kitchens of Copenhagen during the last decade. The share of organic food in public meals was increased to 87% by 2017 without an increase in the costs through a combination of changing tender requirements and working together with the kitchens [9]. Importantly, all food procurement is documented, so it is possible to determine the climate impact of the current diet and implement possible changes to this diet while considering variations between assessment methods. Additionally, by selecting schools with different base conditions, such as pupil composition and dietary requirements, the importance of an individualized approach and the potential impact of differences between schools can also be addressed.

As part of research into the transition of food procurement in public kitchens, the Copenhagen House of Food, the University of Copenhagen, World Resources Institute and Potsdam Institute for Climate Research collaborated on the Climate KIC project "Shifting Urban Diets" led by the EAT Foundation. The project focused on implementing the EAT-Lancet recommendations for a local adaptation of the Planetary Health Diet [17] in the Municipality of Copenhagen, which the Danish Food Based Dietary Guidelines are a local adaptation of. This study originated as part of this collaboration.

Previous studies have reported on the significant climate mitigation potential of transitioning to more plant-based diets [2,3,18–20] and the effects of the choice of LCA methodology on the estimated emissions related to certain food items [11–13]. A previous study calculated the effect on GHG emissions of transitioning from the current average Danish diet to consumption aligned with the national dietary guidelines using two different LCA databases [21]. This study adds to the previous one by using other databases and focusing specifically on institutional kitchens. In the context of institutions, a few studies have investigated food-related GHG emissions in different ways, such as based on specific meal plans [22] and as part of a project aimed at reducing emissions across multiple activities in schools with a focus on student involvement [23]. Finally, a previous Danish study estimated the dietary changes necessary in childcare facilities in Copenhagen to achieve the 25% reduction goal of the municipality [24], which this study complements by focusing on schools.

This study adds to previous research by (i) combining input directly from school kitchens and dialogue with staff with detailed procurement data documenting the actual food purchased over a year, (ii) demonstrating the complexity of this type of assessment by including three different databases of food emission factors compiled using different LCA

methodologies in the context of specific case studies, and (iii) highlighting differences in the baseline food procured by three school kitchens that were selected as cases and potential differences in the effect on emissions of specific transition scenarios at the individual school level.

The objectives were firstly to determine the baseline GHG emissions from food procured by the kitchens of three selected Food Schools and subsequently calculate scenarios for reducing food-related GHG emissions of school kitchens. The calculations were carried out using three databases based on different LCA methodologies to investigate the importance of methodology on the result. The assessment took into account differences between the schools and their local constraints and preferences to showcase how scenarios of this kind cannot be centrally decided upon but rather have to be tailored to conditions in the individual institution. An additional objective was to assess the potential effect of implementing the scenarios on the food expenditure of schools to ensure that changes to the diet would not have negative effects on the budget.

## 2. Methods

### 2.1. Schools Included in the Study

Three Food Schools in Copenhagen were included in this study. Two of the three schools are ordinary public schools, while the third is a smaller public school specifically for pupils with special needs, such as ADHD and autism (Table 1). The schools were chosen for the study based on their willingness to participate. The schools are located in different parts of the city, which affects the composition of the student body. It was required that the schools provided data about their procurement during a year and that they were willing to participate in preparatory interviews and research visits conducted by students. The schools comprise 20% of Copenhagen's total number of Food Schools.

**Table 1.** Schools included in the study and participation in the lunch scheme [25–28].

| School | Pupils (No.) | Internal Employees (No.) | Consuming the School Lunch (%) | Description |
|---|---|---|---|---|
| School A [1] | 724 | 89 | 66 | Ordinary public school. Does not serve pork, and the meat served is halal. |
| School B [2] | 860 | 120 | 60 | Ordinary public school. No specific dietary restrictions. |
| School C [3] | 129 | 62 | 77 | School for children with special needs. Does not serve pork. |

[1] Average of information from 2015 and 2019. [2] Information from 2017. [3] Current levels. Numbers have not changed significantly in recent years.

### 2.2. Assessment of Baseline GHG Emissions from the School Kitchens

2.2.1. Assessment of Baseline Food Procurement

The baseline GHG emissions from the production of food bought by the school kitchens were assessed based on purchasing data obtained from the supplier combined with emission factors from life-cycle assessment (LCA) databases. The baseline assessment was based on procurement data from 2017. For School B, the available data covered the entire year (1 January 2017 to 31 December 2017), whereas for Schools A and C, the data collection ranged from 21 January 2017 to 31 December 2017. Data from the two latter schools were therefore subsequently harmonized to be equivalent to one year by adjusting the purchased amounts of all individual products under the assumption that the type of products purchased during the first 20 days of January did not deviate significantly from those purchased in the rest of the year. All individual food items purchased by the school were mapped to relevant foods or food groups in the LCA databases used. The schools each procured 500–700 different food products during the baseline year, including unprocessed

single-ingredient items such as fruit or eggs, processed single-ingredient items such as flour or spices, and processed multi-ingredient items such as condiments or bread. None of the databases contained every single specific product purchased by the schools, so proxies were used in some cases to map products not included in the specific database. Proxies were selected as much as possible based on the similarity in the type of food product and geographical origin. In most cases, very similar proxies were used, such as using tomato ketchup as a proxy for another processed, tomato-based condiment. For some processed foods, no similar product was available, so they were mapped according to the main ingredient of the product, e.g., candy products being mapped as sugar. For a few products, an obvious proxy did not exist. This was the case for some baking ingredients, such as baking powder or food colorings, and some spices. These were mapped using other products in the same general category, such as the category of "spices". Importantly, these products were used in very small amounts and comprised a very small fraction of the total food procurement, so they did not have a significant impact on the overall result.

After mapping all the food items to corresponding items, categories, or proxies in the databases, the calculation of baseline emissions was conducted by multiplying the procured amount of each food item with the corresponding emission factor from the database. Food items and their related emissions were then sorted into categories, such as beef, dairy, fruit, etc., to enable adjustment on a category level in the scenarios, two of which were based on adjusting the procured amount of the categories beef and meat (Section 2.3).

### 2.2.2. Use of LCA Databases and LCA Methodology

Because of the large number of existing approaches within the family of LCA methodologies, including when applied to food [29], three independent LCA databases of food GHG emission factors were used to present a span of baseline emissions and reduction potentials and evaluate the scale of the differences between them. This is important to showcase how comparability between assessments made with different methods is affected and discuss how the choice of LCA methodology behind the databases affects the outcome of applying emission factors to the cases. The main methodological differences between the databases are (i) whether the database is based on a large array of individual food items or larger food groups, (ii) whether the LCA conducted is attributional or consequential, and for the attributional databases, (iii) whether the allocation in the attributional LCA is carried out according to a bottom-up or top-down approach (Table 2).

**Table 2.** Overview of databases used in the study.

| Database Name | LCA Type | System Boundary | Approx. Food Items/Groups (No.) | Description |
|---|---|---|---|---|
| The University of Copenhagen database of food emission factors (UCPH) | Attributional; bottom-up, town-down, and hybrid data included. | Cradle to regional distribution center (RDC). | 400 | Primarily single ingredients, some processed foods included. |
| The Cool Food Pledge Calculator (CPF) | Attributional; bottom-up. | Cradle to point of purchase by signatory. | 50 | Broader food categories. |
| Den Store Klimadatabase (The Big Climate Database—DSK) | Consequential. | Cradle to retail. | 500 | High level of specificity, many processed foods included. |

(i) Number of food items included: A database consisting of many individual food items allows more precise mapping. However, for foods that are not included and where no directly similar proxies exist, the accuracy of the assessment may be challenged because there are no average food group values to apply to them. A database consisting of fewer, larger food groups has the opposite limitation. The mapping is less precise, but it is less challenging to map niche products because they can be included in a more general category.

(ii) Attributional or consequential LCA: The attributional LCA method is aimed at quantifying what fraction of the studied environmental effects can be attributed to an individual product or process. It attributes these effects through a normative modeling approach, where outputs and inputs to the system processes are allocated by the functional unit [30,31]. A consequential LCA is change-oriented. It is aimed at estimating what the environmental effects will be from a change in a system where products and processes are linked in a market setting [30,32]. Both approaches have strengths and weaknesses, and the choice of approach depends on whether a current or past static system or the future effect of a change or decision is being analyzed [31].

(iii) Allocation in attributional LCA: In attributional LCA, different choices can be made regarding how to allocate the environmental effects to various stages in production, distribution, etc. The main aspect relevant here is the difference between a bottom-up and a top-down approach. A bottom-up approach, or process-based LCA (PLCA), assesses the inputs and outputs at every step in the production of every part or ingredient in the final product, quantified in, e.g., mass [33]. A top-down approach, or economic input–output LCA (EIO-LCA) assesses the environmental effects from a bird's eye view. It is based on economic transactions between different sectors instead of physical inputs and outputs. The economic flows are used as a proxy for modeling the flow of goods and services between sectors, which are then linked to environmental effects [32].

The UCPH database is primarily based on a comprehensive systematic review by Clune et al. [34], supplemented with values from other studies if none were available in the main source. The system boundary is cradle to regional distribution center (RDC), meaning that emission factors obtained from this review include emissions from agricultural production, upstream emissions related to inputs for production (fertilizer, feed, etc.), processing and packaging, and transport up to the point right before the product is distributed to retailers. Values obtained from other studies with other system boundaries were harmonized using estimates of emissions from processing, packaging, transport, and retail [34]; emissions from transport to household by the consumer; and refrigeration and cooking [35]. Three types of attributional LCA approaches were included in the main review used [34]: (i) process-based (bottom-up), (ii) economic input–output (top-down), and (iii) hybrids between these two methods. An assessment of potential bias was made comparing median values for studies using bottom-up and top-down methods in two selected food categories: beef and dairy. These categories were selected due to the high number of relevant studies across the LCA approaches. It was found that there was a high correlation between results from process-based and economic input–output studies [34].

The CFP database was developed as a global initiative by the World Resources Institute. Its purpose is to assist food providers in assessing the climate impact of the food served and provide knowledge needed to diminish this impact [36]. The calculator, based on Poore and Nemecek [2], is used by signatories of the pledge to determine their baseline food-related GHG emissions. The system boundary of this database is comparable to the UCPH database as its scope is cradle to point of purchase by a Cool Food Pledge signatory and therefore excludes emissions at retail and consumer stages. Beyond the farmgate, the CFP calculator includes emissions from transport, processing, packaging, and, unlike the UCPH database, upstream losses along the supply chain [33]. In general, the emission factor estimates for most food groups are higher in the CFP database than in the UCPH and DSK databases. This is likely due to differences in the specific factors included in post-farmgate emissions as well as due to the inclusion of upstream losses in the CFP calculator.

The DSK database was developed by 2.0 LCA Consultants for the Danish think tank Concito [37]. It is made up of very specific food items, a large proportion of which are processed foods, such as preserved foods, prepackaged meals, baked goods, snacks, and beverages. For example, it includes different values for 10 types of frozen pizza but has no values for almonds. It includes no general overarching categories, such as fruit, spices, etc. However, unlike the other two databases, the values in the DSK database are based on a consequential LCA. Emission factors in the database therefore include not only

emissions arising from the individual product but also an estimation of the effect that purchasing a given product has on emissions from related products on the market [38]. The database also includes land use change and indirect land use change, in contrast to the two other databases. Furthermore, economic allocation of emissions from processes resulting in more than one product is used. Therefore, the total emissions from ruminant meats are high and emissions from dairy are lower than in the UCPH and CFP databases because the meat has been assigned a higher economic value than the milk produced. The scope of the database is cradle to retail. However, emissions from the retail stage have a low impact on total emissions, estimated to represent 10–30 g $CO_2$e per kg of product [38].

### 2.3. Scenarios for Reduction

Three scenarios were included in this study and calculated for all schools at implementation levels from 0–100%, assuming a linear implementation (Table 3). These scenarios were built upon scenarios developed by master's students at the University of Copenhagen as part of the course "Climate Solutions". They were developed based on visits to a school and interviews with staff members. Subsequently, scenarios were synthesized from the student-developed scenarios and calculated for all schools.

**Table 3.** Scenarios developed based on the Food School kitchens.

| Name of Scenario | Description (at 100% Implementation) |
|---|---|
| No Beef (NB) | All beef is replaced 1:1 with poultry. |
| Vegetarian (VEG) | All beef, pork, poultry, and fish is replaced 1:1 with dried legumes with a similar protein content. |
| Danish Food Based Dietary Guidelines (DDG) | Procurement is aligned with the official Danish dietary guidelines built upon the Planetary Health Ddiet [17]. |

The three scenarios illustrate different development trajectories with varying levels of ambition. Scenarios that are ambitious beyond what was determined realistic at present by the school were included to showcase the increased potential of GHG reductions from such scenarios, regardless of their real-life relevance for the case schools.

The three scenarios were as follows: (i) The No Beef (NB) scenario, where all beef is replaced 1:1 by poultry. As two of the schools do not serve pork for cultural and religious reasons, this would leave them with poultry and mutton (of which all schools consume very little) as the only meat sources besides fish and seafood. (ii) The Vegetarian (VEG) scenario, where all meat (beef, pork, and poultry) as well as fish and seafood are replaced 1:1 with legumes, specifically dried legumes, such as dried beans, chickpeas, and lentils, as these have a similar protein content by weight. (iii) The Danish Food Based Dietary Guidelines (DDG) scenario, where the entire procurement is aligned with the current official dietary guidelines.

The DDG scenario is more complex than the other two because it is a complete adjustment of the school food to fit the guidelines instead of a replacement of individual food groups. Because the Food Schools do not provide the entire daily food requirements for their pupils, it was not possible to directly use the average daily or weekly amounts from the guidelines as they are recommended based on the daily intake of an adult [39]. Instead, the recommended amounts (in g $day^{-1}$) were converted to percentages (Table 4). These were used to calculate the proportion of individual food groups in the school food, assuming that the total quantity of food consumed would remain at the current level when a scenario was implemented. The adjustment to the dietary guidelines was carried out with individual consideration to conditions at the case schools, e.g., the schools not currently buying pork did not have pork added to their procurement in the DDG scenario. For animal products, added fat, and added sugar, the method was to adjust the purchase to either be at or below the mean percentage in the guidelines. Thus, if a school's purchase of dairy was already less than 16.5% by weight of the total food procured, it remained at its current level. If not, it was adjusted to 16.5%. For vegetable foods, the goal was to adjust the amounts

so procurement of these foods was either at or above the mean percentage. For example, if vegetables already accounted for 18.4% or more of the total food procured by a school, it remained at its current level. If not, it was adjusted to 18.4%. This was carried out to not inadvertently make the scenarios more unhealthy or "green" than the school's current procurement. Grains, the major source of carbohydrates in the diet, was used to sum up to 100% when other food groups had been adjusted according to the principles mentioned above. In the Planetary Health Diet, of which the Danish guidelines are a local adaptation, grains are used in an equivalent manner to add up the total recommended energy intake because no scientific evidence exists supporting an exact percentage intake of grains as long as carbohydrates constitute no more than 60% of the energy intake [17].

**Table 4.** Recommended intake in the Danish Food Based Dietary Guidelines for adults [39] and percentages of total intake amount. Recommended raw intake was used if available in the source.

| Category | Amount (g day$^{-1}$) | of Total (%) |
| --- | --- | --- |
| Whole grains | 390 | 23.9 |
| Potatoes | 100 | 6.1 |
| Vegetables | 300 | 18.4 |
| Fruits | 300 | 18.4 |
| Legumes | 40 | 2.4 |
| Nuts | 46 | 2.8 |
| Dairy | 270 | 16.5 |
| Red meat | 19 | 1.2 |
| Poultry | 38 | 2.3 |
| Fish | 63 | 3.9 |
| Eggs | 15 | 0.9 |
| Added fats | 29 | 1.8 |
| Added sugars | 23 | 1.4 |
| Total | 1633 | 100 |

The schools consider meat an important part of the meal; this is agreed upon by pupils and the kitchen staff. However, many kitchens in Denmark and globally are conscious of the role red meat plays in the climate footprint of the food they serve and are already trying to reduce the amount. As all scenarios are presented at implementation levels of 0–100%, the VEG scenario can showcase various levels of reduction in meat consumption without adjusting other aspects of the diet, e.g., a 50% implementation of the scenario corresponds to a 50% reduction in meat consumption.

As the scenarios in general assume a level of replacement of some food groups with others, e.g., meat replaced by legumes, it was necessary to assess whether this replacement could negatively affect the average macronutrient composition. This was conducted by applying average values for the content of protein, fat, and carbohydrate to individual food groups and calculating the amount of energy obtained from those three sources in the individual scenarios. The data on nutrient content was obtained from a database maintained by the Danish National Food Institute [40]. Due to the considerable number of specific products purchased combined with the average nutritional values used, this assessment can only be seen as indicative and can by no means be used as a certain argument for the health impact of implementing the scenarios. Despite this caveat, it was tentatively concluded that none of the scenarios appeared to have negative effects on the macronutrient composition of the food. All the scenarios seemed to increase the amount of energy coming from carbohydrates but in no cases beyond the 60% threshold that has previously been identified as the maximum intake in a healthy diet [17]. In general, the effect on total intake of protein and fat was negligible, but most scenarios increased the percentage of energy coming from protein slightly and decreased the percentage coming from fat. Importantly, all scenarios reduced the percentage of energy coming from saturated fat (of animal origin). Further information on this supplementary assessment can be found in Table S1.

*2.4. Statistical Analysis and Confidence Intervals of Emission Factors*

An ANOVA test of variance was performed to identify whether there were statistically relevant differences (significance threshold: 0.05) between the results from the three databases and/or between the three schools when implementing the scenarios. Additionally, the confidence intervals for emission factors across the three databases was calculated based on the aggregated data for all schools. Because the databases contain a different number of food categories, ranging from the CFP database using broader categories to the UCPH and DSK databases being more granular in different ways, it was necessary to adjust the emission factors to fit the CFP database to be able to calculate the confidence intervals. This was conducted using the following method: (i) annual amount of food procurement was aggregated into the category of the database with the fewest categories (CFP) and averaged across all three schools, (ii) average annual GHG emissions were calculated with the UCPH and DSK databases aggregated to match the categories of the CFP database, and (iii) average emission factors matching the categories in the CFP database were determined for the three databases. Subsequently the confidence intervals were calculated. These can be found in Table S2.

The type of uncertainty that is displayed in the confidence intervals arises from the methodological choices behind the databases. It can be assumed that the databases utilize some of the same original data, so the variability is a structural uncertainty as described by Solazzo et al. [41].

*2.5. Implications of Scenarios on Food Expenditure*

The economic effects of implementing the scenarios were assessed directly based on the expenditure of the kitchens in the baseline year as the purchasing data obtained from the supplier for each school also provided price information. Expenditure on different food groups was calculated and adjusted with the same percentage as consumption of the food group affected by the scenario. This means that no changes in the prices of individual commodities were assumed; the only effect on the economy of the school arising from the scenarios was due to the increasing procurement of some food items and the reduction in procurement of others. Additionally, it was assumed that the schools would continue to purchase the same products within the product groups as was the case in the procurement data, i.e., if a school in the baseline bought more expensive meat or cheaper dairy than the average, this was assumed to also be the case in the dietary scenarios.

## 3. Results

*3.1. Baseline Procurement and Related Emissions*

Baseline food procurement (t year$^{-1}$) and related emissions (t CO$_2$e) were calculated using the UCPH, CFP, and DSK databases for all three schools (Table 5). Schools A and B purchased almost the same amount of food per year (21.2 and 21.1 t year$^{-1}$). It is notable that even though Schools A and B were 6–7 times larger in terms of pupil numbers than School C, the total procurement of food in School C was only 13–14% lower than that of the other schools.

It is possible that School C's high procurement was associated with more waste during the baseline year due to the special circumstances of its pupils. The school reported that children with autism and ADHD have lower appetites and are selective when it comes to food. Additionally, the pupils are easily distracted and overstimulated at mealtimes, often leading to more leftovers. Another possible factor contributing to the high amount of food purchased by School C was that School C also reports serving snacks (most often fruit) in addition to the school lunch. In support of this, the amount of fruit purchased by School C was 78–124% higher than the amount purchased by the other schools.

**Table 5.** Total food procured by the schools A, B, and C in a year and the related GHG emissions in total for the entire school year, and per ton of food procured.

| | Database | School A | School B | School C |
|---|---|---|---|---|
| Total food procurement (t) | | 21.2 | 21.1 | 18.3 |
| GHG emissions (t CO$_2$e) | UCPH | 55.0 | 65.3 | 52.0 |
| | DSK | 63.2 | 66.0 | 56.4 |
| | CFP | 77.1 | 94.8 | 74.7 |
| Emissions per unit of food (t CO$_2$e t$^{-1}$) | UCPH | 2.59 | 3.10 | 2.84 |
| | DSK | 2.98 | 3.14 | 3.08 |
| | CFP | 3.63 | 4.50 | 4.07 |

Table 5 shows the differences between the databases and between the baseline emissions per unit of food that the schools procure. Emissions from the CFP database were 40–45% higher than those found when using the UCPH database, and results from the DSK database fell in between those, generally with results closer to the UCPH database. Differences between baseline emissions per unit of food calculated by the databases were statistically significant in an ANOVA test of variance (*p*-value 0.005), while the differences between the schools' individual baseline emissions per unit of food were not (*p*-value 0.7). Thus, the differences in the methodology used for calculation of the database emission factors were large enough to be statistically significant when applied to the three case schools, which have different procurement practices. The differences between the schools regarding the food they bought (Figure 1) were not large enough to obscure the effects of the database methodology on the calculated GHG emissions. Additionally, the choice of database led to markedly different GHG emission profiles of the procured food (Figure 2).

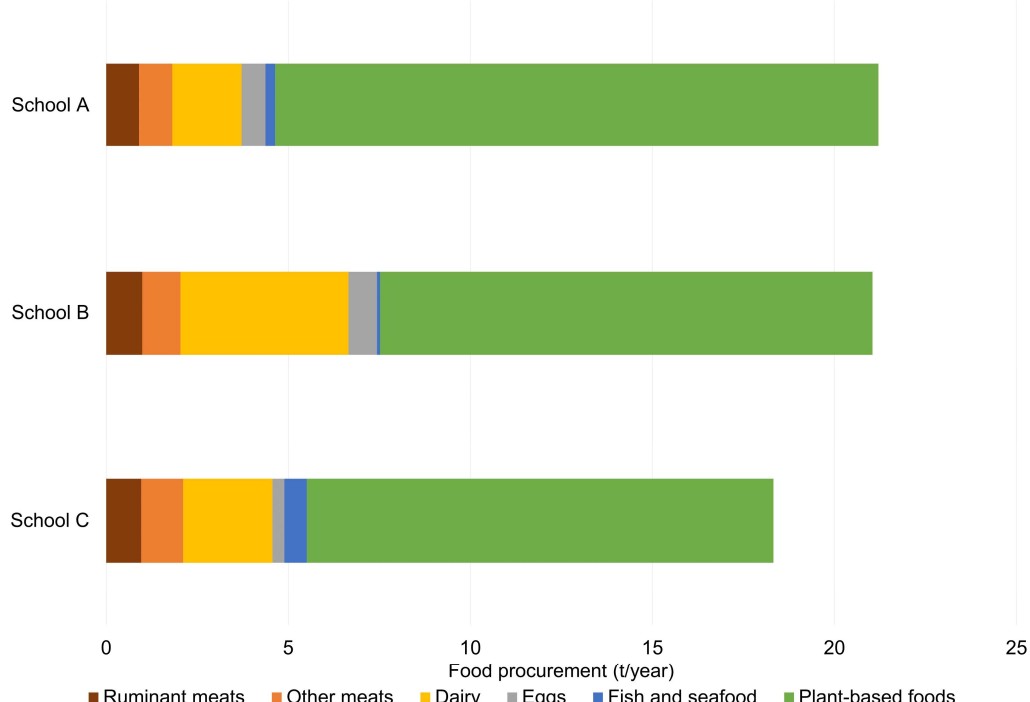

**Figure 1.** Procurement of different food groups in the three school kitchens.

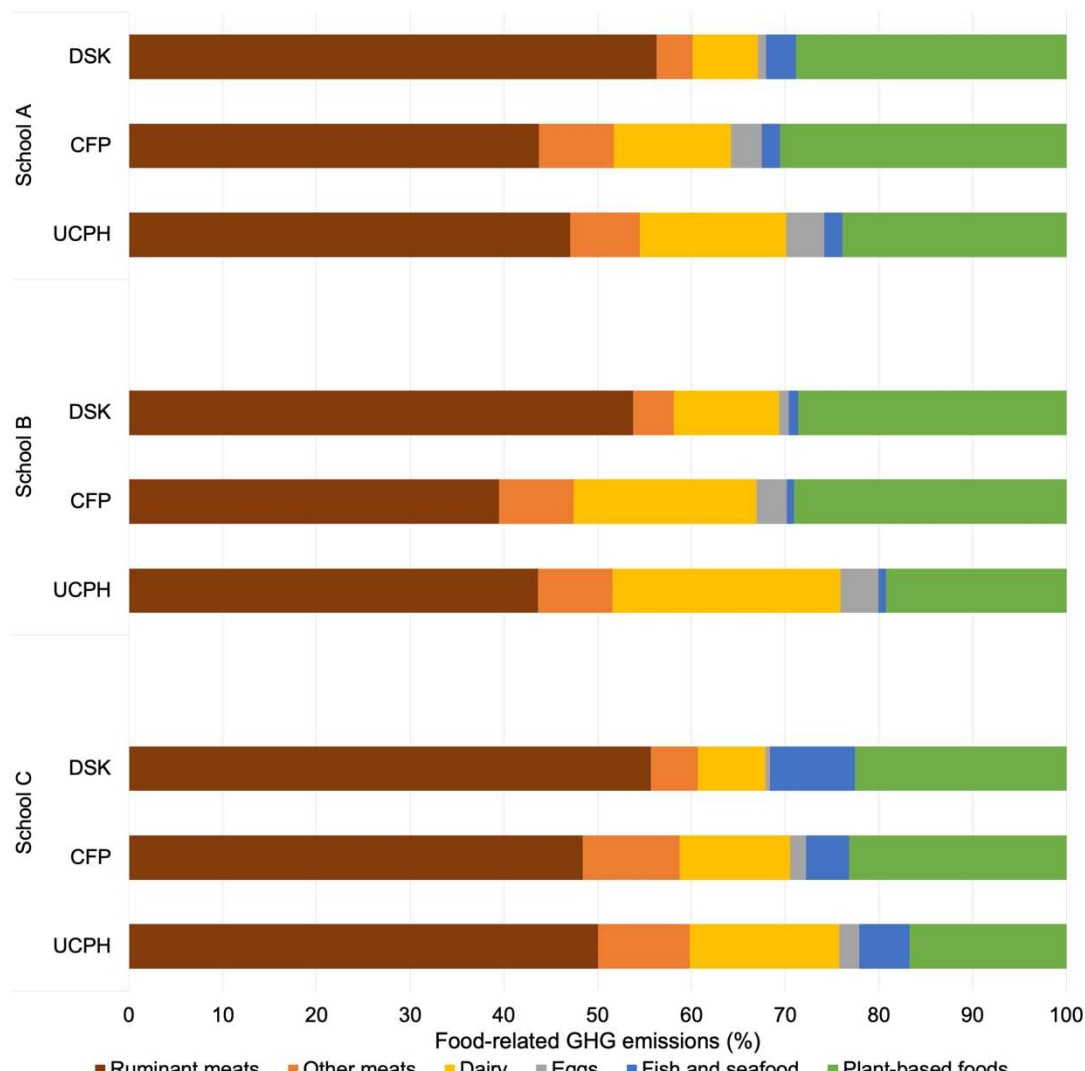

**Figure 2.** GHG emissions from different food groups in the school kitchens.

The significant differences identified between the GHG emissions when calculated using the three databases resulted from variation in the methodology. The databases have multiple differences, and it was not possible to pinpoint the discrete methodological choices in the databases that led to them. The differences were a composite of whether the LCA was attributional or consequential, the number of food categories included, and the exact system boundary used. Variability in the emission factors for the food categories resulted in confidence intervals of +/−10 to +/−122%, with emission factors for wine, peas, and miscellaneous vegetables displaying the lowest uncertainty and miscellaneous stimulants and spices, coffee, and onions and leeks showing the highest uncertainty (Table S2).

When looking at the databases, the most evident overall difference was between the CFP database and the other two because the CFP database generally resulted in higher emission estimates across most food categories. In absolute terms, the highest contributors to the higher total emissions when using the CFP database were animal-based foods, especially ruminant meat, because these foods are so carbon intensive. The emissions from ruminant meat was so high that even though the uncertainty in the emission factor between the databases was among the lowest, it added up to a large difference when multiplied with the procured amount. On average, across the schools, differences in emissions from ruminant meat accounted for 36% of the total difference between the CFP (the highest) and UCPH (the lowest) database results while accounting for less than 5% of procurement on average. In relative terms, an important difference between the databases is the limited

number of categories in the CFP database relative to the other two. For example, the CFP database has a miscellaneous category called "stimulants and spices (misc.)", which has a much higher emission factor than the aggregate of the corresponding categories in the UCPH and DSK databases. The relative uncertainty between the databases was highest in this category, with a confidence interval of +/−122% (Table S2).

Other differences between the databases include how emissions from ruminant meat and dairy are considered relative to each other as well as emissions from animal-based foods versus plant-based foods. In the DSK database, ruminant meat plays a larger role and dairy a smaller role. Conversely, the UCPH database allocates lower levels of emissions to plant-based foods relative to total emissions than either of the other databases, making the contribution from all animal products appear higher (Figure 2). The distribution of emissions within the overarching category of animal products is especially important because these products are generally associated with higher emissions per unit of food. Ruminant meat was the highest emitting individual food category, followed by dairy, across all schools and all databases (Figure 2) despite being consumed in small quantities (Figure 1), while the reverse was true for plant-based foods. The differences between the databases affected the estimated ranges of emissions differently in the three schools because of variations in procurement. The allocation of emissions to beef versus dairy especially affected School B, which had a high procurement of dairy. For this school, the estimated emissions per unit of food using the DSK database were only 1% higher than the estimate using the UCPH database. For the two other schools, this difference was 8–15%. Because dairy is considered less responsible for emissions relative to beef in the DSK database than in the UCPH database, the resulting emissions from School B appeared lower when using the DSK database. Importantly, the differences between the schools were not statistically significant.

### 3.2. Baseline Food Expenditure

The baseline food expenditure divided between the main food groups was assessed for the three schools, and in general, their spending patterns were similar (Figure 3). The schools spent 39–49% of their food budget on animal products, which accounted for 22–36% of their total procurement by weight (Figure 1). Ruminant meat was the most expensive food item purchased by the schools, with 12–13% of the food budget going to beef and mutton, though these items only accounted for 4–5% of the schools' procurement by weight. Relative to the amount (kg year$^{-1}$) of food purchased, School A had the highest expenditure, with spending per purchased kg being 14 and 20% higher than Schools C and B, respectively. This was the case even though School A had the lowest relative procurement of animal products compared to the other two schools. The reason for this can be found in the differences between the types of products within each category the schools purchased. School A purchased animal products that were on average 45 and 53% more expensive per kg than those purchased by Schools C and B, respectively, and plant-based foods that were slightly more expensive per kg than those purchased by the other two schools. Thus, even though the schools all used the same supplier and were all subject to the same municipal regulations, there was still room for maneuvering within the budget according to conditions on the individual school with regard to choosing cheaper or more expensive options within each food category from the supplier. For School A, the main driver behind choosing more expensive animal products was most likely a preference for halal slaughtered meat.

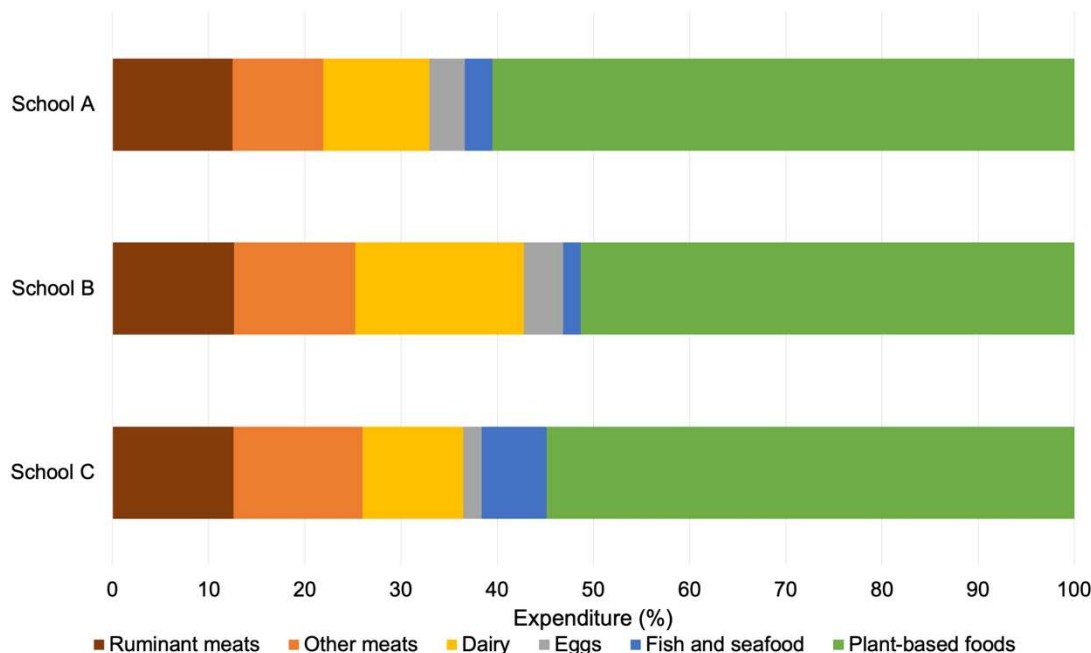

**Figure 3.** Total food expenditure of the included school kitchens.

### 3.3. Potential Emission Reductions from Implementation of Scenarios

The implementation of all three scenarios led to reductions in food-related GHG emissions relative to the degree of implementation (Figure 4), regardless of the database used. For all schools, the NB scenario had the lowest reduction potential and the VEG scenario had the highest, though there were overlaps between the scenarios due to differences between the LCA databases used. In general, the results for emission reductions at full implementation of the scenarios followed the tendency seen in the baselines, where the UCPH database resulted in the lowest emissions and the CFP database the highest from baseline procurement (Table 5). The differences identified between the databases were statistically significant across all three scenarios, whereas the differences between the schools were not (Table 6). Thus, as for the baseline, the methodological choices underlying the database emission factors varied enough to significantly affect the result when applied to the three schools with different procurement practices. Although two of the scenarios (NB and VEG) were based directly on the existing procurement situation and only adjusted specific food categories, the schools were not different enough to override the effect of the variations between the databases. However, the *p*-values of these simpler scenarios were higher than for the DDG scenario, indicating that a confounding effect of the differences between the schools did exist. This was especially the case for the VEG scenario (*p*-value = 0.047), which was close to the 0.05 threshold, indicating that differences between the schools when it came to meat procurement relative to, e.g., other high-emission foods such as dairy, made the differences between the databases less clear than was the case for the baseline (*p*-value = 0.005) or the other two scenarios (Table 6). In the DDG scenario, differences between the schools were almost eliminated due to the adjustment of the whole diet, which meant that the differences between the databases were comparatively clearer.

Though the UCPH database generally found the lowest emissions, in a few cases, the DSK database showed the lowest values, such as at high implementation levels of the NB and VEG scenarios for School C (70 and 100%, respectively) and School A (80 and 100%, respectively). For School B, this was the case for all and almost all implementation levels of the NB and VEG scenarios, respectively. This can be explained by differences in allocation of emissions from cattle production systems between meat and dairy. As the DSK database weighs emissions from meat higher relative to dairy than any of the other databases, implementation of the NB and VEG scenarios calculated with the DSK database

resulted in a larger reduction in food-related emissions from School B, which had a higher dairy consumption, than when calculated with the other two databases.

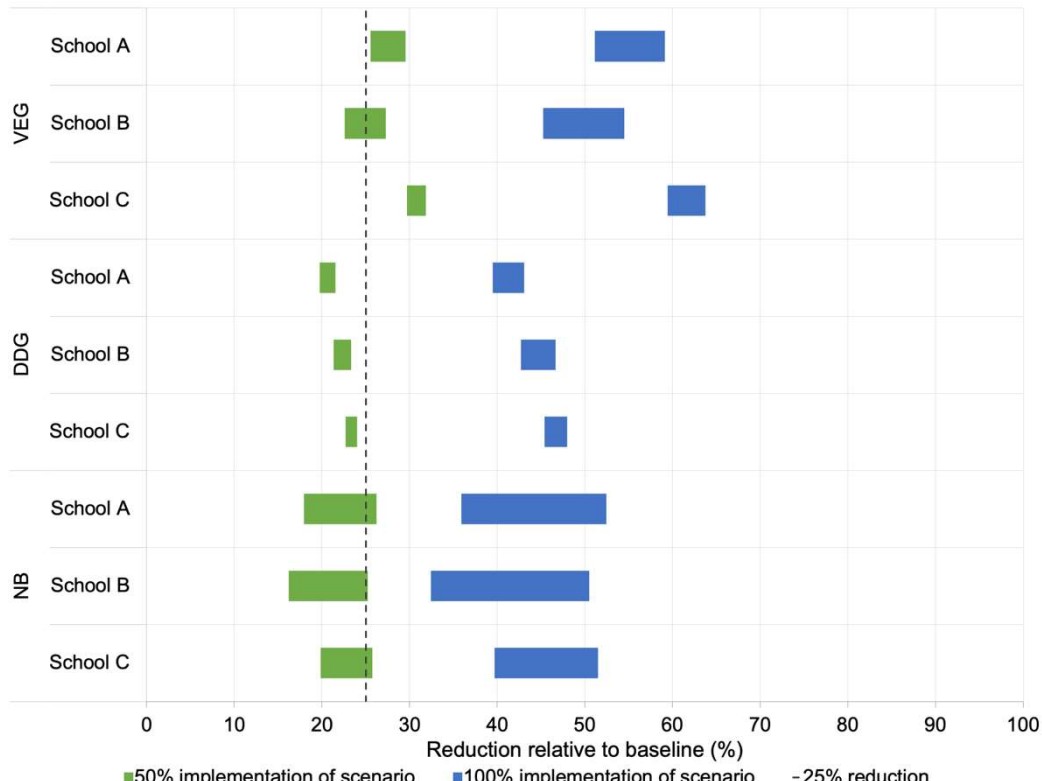

**Figure 4.** Reductions in emissions (%) of scenarios relative to baseline of 50 and 100% implementation of scenarios. The municipality of Copenhagen's goal of 25% reduction in emissions from public kitchens by 2025 is shown for reference.

**Table 6.** ANOVA (analysis of variance) between databases and schools.

|  | NB | VEG | DDG |
|---|---|---|---|
| Between databases (*p*-value) | 0.0049 | 0.047 | <0.001 |
| Between schools (*p*-value) | 0.71 | 0.34 | 0.83 |

All three scenarios for all schools did lead to larger reductions than the municipality's current target of 25% at higher implementation levels (Figure 4). Full implementation of the VEG scenario led to reductions of 45–64%, while the DDG scenario led to reductions of 39–48% and the NB scenario to reductions of 32–52%. Thus, by simply adhering to the national dietary guidelines, emissions from school food can be reduced by approximately 40%. When comparing the effect of the scenarios, it is important to be aware that the DDG scenario is deceptively ambitious with regard to meat consumption as the current dietary guidelines include much less meat than current diets. Full implementation of the DDG scenario would, in fact, lead to reductions in meat consumption of 60–71% relative to baseline levels.

The effect of the difference between the schools with regard to dairy consumption was evident in the VEG scenario, and it is interesting to consider this even though differences between the schools were not statistically significant because it shows that a whole-diet approach is preferable to simply adjusting individual components. In the VEG scenario, the potential reduction in emissions for School B was lower than for the other two schools at both 50 and 100% implementation (Figure 4). This is due to the higher dairy consumption in School B. The NB scenario had the largest span of possible reduction levels due to differences between the databases with regard to dairy and beef. This scenario only reduced

intake of beef, which highlighted the differences between the databases when it comes to allocation between beef and dairy. The relative uncertainty of the emission factor for beef was among the lowest found across the food categories (Table S2), but because of the high emissions related to ruminant meat production, this relatively low uncertainty nonetheless resulted in large variation when multiplied with the amount procured by the schools.

*3.4. Changes in Food Expenditure from Implementation of Scenarios*

All the scenarios led to reduced food expenditure for all schools, with the VEG scenario generally producing the highest relative savings (13–18%) and the NB scenario the lowest (1–4%). The DDG scenario resulted in savings of 5–13%. This was due to the current disproportionate amount spent on meat relative to the size of procurement (Figures 1 and 3). For the DDG scenario, and especially the VEG scenario, relevant savings were also possible at 50% implementation levels. In the VEG scenario at 50% implementation—equal to a halving of meat procurement—a reduction on food expenditure of 6–9% was achieved, equal to about EUR 5200–7100 per year. The NB scenario resulted in such limited savings for the schools because beef in this scenario was replaced with poultry. The small reduction achieved was due to differences in the price of the beef and poultry the schools purchased in the baseline.

## 4. Discussion

The three scenarios, NB, DDG, and VEG, present different suggestions for how an institution can achieve a sizable reduction in GHG emissions from food procurement and highlights how important it is to be aware of the LCA methodology used for calculation of food-related GHG emissions and pathways towards reduction.

The NB, DDG, and VEG scenarios led to GHG emission reductions of 32–52%, 39–48%, and 45–64%, respectively, depending on the school and database used. These findings are comparable to results from other studies exploring the potential of more plant-based diets for climate change mitigation. These results include GHG emission reductions of 61% by adoption of the Planetary Health Diet [18], 58–86% by transforming the Danish agricultural sector to match the Planetary Health Diet [42], 56% by adopting a flexitarian diet [3], 29–70% by various diets with lowered meat intake [43], and 49% by adopting a vegetarian diet [2].

The three databases used in this study produced estimates for GHG emissions that were different at a 0.05 significance level for all the schools, both at the baselines and in the scenarios due to the different LCA methods used in the databases. Implementation of the NB scenario produced the most variable result of the three scenarios, even though the uncertainty in the emission factor for beef was found to be relatively low. This is due to the high emissions in absolute terms of ruminant meat, which amplifies even a small uncertainty. The variation identified between the databases were, of course, present in all scenarios but was less pronounced when multiple products were included, though it was still statistically significant in the other scenarios as well. A previous study comparing dietary transition scenarios similar to the DDG concluded that the choice of methodology is important for the calculated effect of the transition and that this is relevant knowledge when developing dietary guidelines [21]. Our study adds more database comparisons supporting this conclusion and finds that this difference is pronounced in simpler scenarios that do not address the whole diet and that it also holds true when applied to case studies with different baseline conditions. Efforts are being made to reach some level of consensus when it comes to the allocation issues in LCA of meat [44], but no unified approach exists as of yet. The effect of the LCA methodology on the results of the dietary scenarios highlights the importance of ensuring that the assumptions and approaches underlying emission factors are known and understood. This is important knowledge for restaurants, cafés, and canteens that want to calculate their emissions or use carbon footprinting of their menus as a communication tool. Both the CFP and the DSK databases are publicly available and both are currently being used by cities, municipalities, and other institutions, so a need for

awareness about the importance of the choice of database is evident. This is important if comparisons are intended across institutions but also for the individual institution using carbon footprinting as a starting point for implementing initiatives to reduce emissions because the choice of database evidently affects the view of the proportion of emissions coming from different food groups.

Reducing the carbon footprint of diets is a matter of urgency due to the importance of food systems for global GHG emissions. Due to their size and reach, public kitchens can play a significant role in making this happen. However, it is important for the kitchens—and their consumers—to trust the calculations and estimates they obtain. Presenting the carbon footprint of a meal in a canteen or using the calculated reduction in emissions from implementing a change as a tool for communication can make it seem as if these values are undeniable facts. Clearly, when databases result in significantly different levels of GHG emissions, they are not. This is an important issue to be aware of and continue sharing with food professionals who are keen to work with the carbon footprint of their meals. Consumers have been found to be aware of the general order of importance of different food groups for the carbon footprint of diets, but they lack knowledge about the magnitude of the difference between them [45]. For this reason, consumers cannot be expected to be able to identify the uncertainties related to information presented to them about the climate impact of food and diets.

Importantly, the narrowest span in potential emissions estimated with the databases was seen when using a whole-diet approach, as in the DDG scenario. This supports an approach to reduction in food-related GHG emissions that addresses the whole diet instead of targets such as aiming for a set reduction in consumption of specific foods. The most certain approach is also in some ways the simplest and most justifiable: adjust meals to fit the Planetary Health Diet or a local adaptation, such as the Danish Food-Based Dietary Guidelines. The Planetary Health Diet assesses all diet components and is aimed at achieving a diet that is healthy without leading to environmental effects that exceed the planetary boundaries [17]. Most current national dietary guidelines do not take sustainability into account at present, and suggestions have been made for how to create dietary guidelines that combine human health with climate and land use impacts [46]. However, the Planetary Health Diet—and its local adaptations by extension—remains a solid approach for achieving this. Addressing the whole diet makes the DDG scenario better equipped to ensure that other components of the diet do not change towards a more carbon-intensive or less healthy direction after a single aspect has been addressed. This could be the case in the other two scenarios where, for example, dairy consumption could potentially increase while meat decreases. Consideration of these trade-offs necessitates a holistic approach that addresses the larger framework. Among the scenarios, this makes the DDG scenario the most useful for real-life implementation in schools, institutions, or canteens, and scenarios based on the Planetary Health Diet or a local adaptation can be directly implemented in kitchens globally. The municipality of Copenhagen is already acting on this by implementing meal plans living up to the standards in the dietary guidelines on some Food Schools [47].

Given the analyses undertaken here, it is likely that all scenarios can have a positive effect on the nutritional profile of schools' meal plans because they all reduce the amount of red meat in the diet. As previously mentioned, a supporting assessment of the effect of the scenarios on the nutritional profile of procurement has been carried out as a supplement to this study. This assessment is not precise enough to be usable for making conclusions on the specific nutritional composition of the food because it is based on average values for the nutritional content of individual foods instead of exact individual products purchased by the school. It is, however, possible to see a trend, specifically with regard to the percentage of energy coming from fat of animal origin. Animal products are the dominant source of saturated fat, which should be limited in the diet. As all scenarios reduce the intake of animal fat to align better with the recommended levels (Table S1), the health profile of the meals can be assumed to improve at least in this regard.

Additionally, most daily meals are consumed outside of the school—and outside public kitchens in general—so an important additional benefit of a healthy and climate-friendly meal plan in schools is the potential ripple effect such a change can have on the diet and preferences of the general population. It has been demonstrated that people form their adult dietary habits based on the food and cooking practices they are exposed to as children [14]. Furthermore, it has previously been found in a Danish context that schools can be important facilitators in the formation of healthier dietary habits by acting as normative institutions [48]. This puts public school kitchens in a unique position. They are ideally situated for influencing the dietary habits of children and their families in a positive way while also being under public control. School kitchens can thus become catalysts for a large-scale systemic shift in dietary habits and the related GHG emissions, and large benefits could be reaped by ensuring that policies and food procurement practices support this. However, as previously mentioned, it is important that institutions that are interested in calculating and tracking their food-related emissions are aware of the impact of the method they use for the assessment, regardless of whether they intend to use it in communication with users, to generate support for a change in meal plans, or for public communication. It is paramount to not consider an LCA-based result as a fact. This is not an understanding that can be expected of casual users of publicly available databases or consumers.

*Limitations*

The study is limited by being based on individual schools in only one city. The schools are concrete examples of public kitchens that are used to showcase differences in calculated GHG emissions from food depending on the LCA methodology and differences in the potential for implementing initiatives to reduce GHG emissions from food procurement that arises from local conditions and preferences, so the results are not directly transferable to other kitchens in other institutions.

Additionally, the schools, kitchens, and meals plans have potentially changed since the data were collected, so the study does not provide directly usable food-related climate change mitigation strategies that can be pursued by the relevant procurement officers or school chefs. Importantly, because the schools used as case studies are designated Food Schools, at the time of data collection, they were already ambitious with regard to food education and the involvement of students as well focusing on producing food onsite with fresh produce. Due to the preexisting focus on organic food, the schools were also used to weighing highly the sustainability aspects of their food procurement. Many other schools and other institutions, both nationally and globally, will have further to go at present than these schools did at the time of data collection. Importantly, the power of the differences between LCA methods to impact the result and the effect of individualized adaptations and implementation choices has not changed since the data were collected.

## 5. Conclusions

By analyzing the implementation of three dietary scenarios in three case schools, this study has documented that large reductions in food-related GHG emissions are possible through sustainable food procurement for public kitchens. With full implementation of the scenarios, GHG emission reductions were in the range of 32–64% depending on the scenario, the LCA database used, and the baseline meal plan at the individual school. The largest reduction was found with the vegetarian scenario, followed by the national dietary guidelines scenario and the scenario excluding beef. The three databases used for the calculations produced differences in the estimates of GHG emissions that were statistically significant. The largest variation between the databases was found in the scenario that only addressed beef consumption and the smallest variation in the scenario that adjusted the whole diet to levels recommended in the national dietary guidelines. To achieve reductions in food-related GHG emissions that are more certain across calculation methods while also ensuring a healthy, balanced diet, using a whole-diet approach is recommended. Addi-

tionally, basic knowledge about the impact of the chosen LCA method on the results and caution when communicating these results is crucial for kitchens conducting an assessment of their emissions using the available databases.

**Supplementary Materials:** The following supporting information can be downloaded at: https://www.mdpi.com/article/10.3390/su151713002/s1, Table S1: Overview over nutritional composition of the scenarios; Table S2: Average emission factors across databases and confidence intervals.

**Author Contributions:** Conceptualization, C.B.H. and M.L.; methodology, C.B.H., M.L. and A.A.P.; formal analysis, A.A.P.; investigation, J.B.A., F.D., M.S.D., C.F., K.H., S.K., A.M.L., J.P.M., M.M., S.F.M., S.Y.S., K.-L.P.T., I.T., B.V. and F.Z.; writing—original draft preparation, A.A.P., J.B.A., F.D., M.S.D., C.F., K.H., S.K., A.M.L., J.P.M., M.M., S.F.M., S.Y.S., K.-L.P.T., I.T., B.V., F.Z. and M.L.; writing—review and editing, C.B.H. and A.A.P.; visualization, A.A.P. All authors have read and agreed to the published version of the manuscript.

**Funding:** This research received no external funding.

**Institutional Review Board Statement:** Not applicable.

**Informed Consent Statement:** Not applicable.

**Data Availability Statement:** Excel sheet with calculations and scenario development can be obtained from the authors upon request.

**Acknowledgments:** The authors would like to thank the Copenhagen food schools for providing access to their data and contributing to student projects.

**Conflicts of Interest:** The authors declare no conflict of interest.

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
