# Peer review of "Scenarios for Reducing Greenhouse Gas Emissions from Food Procurement for Public School Kitchens in Copenhagen"

_sustainability, doi:10.3390/su151713002_

Round 1
Reviewer 1 Report
Review
The article has well-defined objectives, the methodology is well explained and references are included.
P. 9. The Figure does not have a number and title. It is important to place Numbering (Figure x), Title and below the Source.
It is interesting that the authors add a little more information about food production in Denmark, for example, the origin of the products, that is, if the beef is local, where it is imported from, if the vegetables are local (from the country). It is an interesting point to be added with the aim of drawing more attention from readers who come to know production and consumption. This helps the reader understand where research is being carried out, in what context, etc.
In conclusion it is always important to add research limitations, suggestions for future research and sometimes implications for public policy.
Author Response
We have checked figures and their naming, titles and numbering.
We considered your suggestion to add more information about food production in Denmark, but decided against it. It is indeed interesting, but the information about where the food originates from is 1) Not necessarily available in the procurement data we obtained from the schools, and 2) not relevant on such a detailed level when it comes to assessing the climate impact. This is simply because such precision does not exist in the LCA databases, which are not differentiated at a national level. Thus, we felt that adding this information to e.g. to introduction would not support the method and results in the paper.
We added limitations as a part of the discussions.
Reviewer 2 Report
Comments
1. The manuscript is well-organized and well-written.
2. It is better to add a graphical abstract to the study.
3. The title of Figure 1 is not well-placed.
Author Response
2. We are considering making a graphic abstract. We agree it is beneficial.
3. We have checked the formatting of the paper to make sure titles are well-placed.
Reviewer 3 Report
The research topic is interesting, but some specific methods or data need to be provided, and the project needs a large amount of data to be investigated in order to obtain a reliable result.
My comments related to improving the article are as follows:
Point 1: Why do you choose this topic for research? Are similar studies conducted by predecessors?
Point 2: in line 29, What is global food system ? Please express it clearly.
Point 3: In line 29 - 30, is there really such a large proportion of livestock livestock farming?
Point 4: All tables need to be standard three-line tables.
Point 5: It is well known that automobile exhaust worsens the greenhouse effect. What is the specific significance of this study?
Point 6: In line 25, Keywords should be bold.
Point 7: In line 116, What are the reasons or characteristics for choosing these schools?
Point 8: Please check on your reference format carefully and modify it. Such as reference 3, 6, 20 - 24 and so on.
Author Response
Point 1: This has been added to the introduction.
Point 2: The global food system is a commonly used term, meaning the production, processing, distribution etc. of food, which we do not consider it necessary to define in this context.
Point 3: It is broadly agreed upon that livestock farming is a major contributor to global warming, and it has been quantified in many influential papers, see for example Poore and Nemecek (2018).
Point 4: We are not familiar with what standard three line tables are, and we do not understand why the table can not be mad in a way that matches the data.
Point 5: We do not understand what the relevance is of automobile exhaust is to this paper. It is also well-known that food production is a major contributor to greenhouse gas emissions, and quantification of this, as well as determining the options for reducing it, is an ongoing field of study.
Point 6: We will adhere to the Sustainability format on this.
Point 7: This has been added.
Point 8: We have checked the references, updated some and gone through the formating.
Reviewer 4 Report
The topic undertaken by the authors is very interesting and important from many points of view. However, the article contains some errors and shortcomings that should be taken into account before publication.
1. It would be worth adding to the keywords "reducing greenhouse gas emissions".
2. There is no review of the current literature in the area of the analyzed topic in the manuscript; of course, some aspects are mentioned in the Introduction section, but it's not very extensive. There is no theoretical basis - and almost all of the Introduction refers only to Copenhagen. How will this affect other cities?
3. Do the conducted research and analysis of the results presented in the manuscript fill any gaps in the literature? What is their originality and novelty? After a short Introduction section and no Literature Review section, it's hard to tell.
4. On what basis and according to what criteria were the 3 schools selected for the research?
5. In subsection 2.1. under table 1, the authors specify the periods to which the data contained in the table refer. And first of all - these are different periods - so how can it be compared? And secondly - in subsection 2.2.1, the authors write that they will carry out analyzes on data from 2017 - so how does it relate to the periods presented in table 1? And thirdly -the data described in subsection 2.2.1 is not very up-to-date - from over 5 years ago - has nothing changed since then?
6. The food procurement analysis procedure described in subsection 2.2.1 is not very clear.
7. In general, the entire methodology is not described in an understandable and clear way. There is also no indication of research problems.
8. The obtained results were based on simple calculations, there are no more advanced statistical methods.
9. On what basis have the amounts presented in subsection 3.4 been calculated?
10. There is no indication in the Conclusions section of research limitations, as well as the possibility of using the obtained results - by the city authorities? commune? school?
11. Very poor references - there are too few of them, some are in Danish, so I can't even assess what they refer to and whether they are thematically related to the article. Moreover, there are no current references (although the topic is very current), only a few items from 2021. The authors should definitely improve this part of their manuscript
Author Response
1. Keyword added.
2. More literature has been added to frame the issue in terms of 1) effect of shifting to more plant-based diets, 2) that shift in schools and 3) the importance of LCA methodology in food assessment. The introduction has been updated, to clarify that an important part of the paper is to determine the GHG emissions from three schools used as case studies, based on actual procurement data and not on meal plans etc., and investigate the results from three different databases. Relevance broadly for other schools and institutions has been added in the discussion as well.
3. This has been clarified in the the introduction.
4. This has been added to the methodology.
5. The table referred to is only providing background information about the schools, the analysis is conducted based on actual procurement data. This has been clarified. It has also been clarified that the schools are case studies and that they were generally ahead of most schools, both in Denmark and globally, when it came to considering sustainability in food procurement. For this reason, it does not change the relevance of the scenarios that the data is not very recent. It would be relevant to conduct a follow up to see where the schools are now.
6. The explanation of the analysis has been elaborated.
7. Research problems are located at the end of the introduction, formulated as objectives. The method has been checked and elaborated where it was deemed necessary.
8. Results from and ANOVA test have been added.
9. In the methods a section describes how the expenditure changes were calculated.
10. Limitations have been added in the discussion. Relevance outside of the schools has been added in both introduction and discussion.
11. References have been updated and improved. Danish references can not be completely avoided, but they are limited and are primarily background information about the schools, the municiality's strategy and the Danish database used (Concito). Additionally, some studies remain highly relevant for this topic, regardless of the years that have passed since their publication, e.g. Poore and Nemecek (2018), which is a hugely influential paper that one of the databases used is even based on.
Reviewer 5 Report
Manuscript ID: sustainability-2340751
Title: Scenarios for reducing the greenhouse gas emissions from food procurement of public school kitchens in Copenhagen
The manuscript needs some modifications so that it could be better than before. It would be helpful if the authors would consider the following points:
- Improve the introduction section, providing the problem. Please highlight the advance of the study in Introduction. Please explain the development and creative work. The literature review should be carefully considered.
- The discussion section needs strengthening. The discussion should add the limitations of the study and future directions, as well as discuss the results and explain the practical implications of the work. The authors should discuss the results clearly and should provide relevant information.
- References used should be updated, where it is noted (no reference in 2022 and 2023)
- In each page: "Sustainability 2021, 13, x FOR PEER REVIEW " , update to 2023
- Line 20: Add "the" before "scenario"
- Line 22: Add "with" before " the lowest"
- Line 53: Add "," after " societies"
- Lines 79-81: Rewrite this sentence
- Lines 100-101: Change " different LCA methodology" to " different LCA methodologies"
- Line 127: Change "has" to "have"
- Line 135: Add "," after "cases"
- Line 142: Delete "had"
- Line 149: Change " Whether " to " whether "
- In Table 2: The abbreviation can be written next to the full name in the first column "Database name ". Thus, the second column is deleted
- Lines 204, 416: Change " food related" to " food-related"
- Line 217: Change " such a" to "such as"
- Line 237: "part the course" to "part of the course"
- Line 237: Change " “Climate Solutions.”" to " “Climate Solutions”."
- Line 237: Add "," after " Subsequently"
- Line 241: Add "an" before " objection "
- Line 245: Change "is" to "are"
- Line 275: Change " carbohydrate" to " carbohydrates"
- Line 228: subscript " CO2 "
- In Table 3: The abbreviation can be written next to the full name in the first column " Name of scenario". Thus, the second column is deleted
- Line 315: Change "is" to "are" because "changes"
- Line 325: Change "School A and B" to "Schools A and B"
- Lines 338-339: Rewrite this sentence
- Line 429: Add "the" before " allocation"
- Line 498: Change " leave " to " leaves" because "This"
- Line 529: Change " Animal products is " to " Animal products are"
- Lines 537-538: Rewrite this sentence
- It is preferable to shorten the "5. Conclusions"
Author Response
The introduction has been updated, to clarify that an important part of the paper is to determine the GHG emissions from three schools used as case studies, based on actual procurement data and not on meal plans etc., and investigate the results from three different databases. More literature has been added to frame the issue in terms of 1) effect of shifting to more plant-based diets, 2) that shift in schools and 3) the importance of LCA methodology in food assessment.
The discussion has been adjusted so that the importance of different LCA methods is more evident, and the relevance for practical implications is mentioned. Limitiations have also been added.
Several referances have been added and updated some, so they more are recent (from 2022). Additionally, some studies remain highly relevant for this topic, regardless of the years that have passed since their publication, e.g. Poore and Nemecek (2018), which is a hugely influential paper that one of the databases used is even based on.
Grammatical errors have been corrected.
Round 2
Reviewer 3 Report
The author has made a lot of revisions, but there are still some minor problems that need further revision.
Point 1: Table 2, Table 4 and Table 5 should be a standard three-line table.
Point 2: Please check on your reference format carefully and modify it. Such as reference 4,8, 14, 21-25, 32-36, and 39.
Author Response
Point 1: We have reformatted the tables that were not three-line. However, in table 5, we have added a slight, grey coloring to ensure readability.
Point 2: We have checked the reference format again, and it seems the issue largely is with citation of reports of various kinds. There is no mention of citation of reports in the Sustainability guidelines (https://www.mdpi.com/journal/sustainability/instructions), so we have cited these as books. We have referred to the organization as the author in cases where this is generally recommended, e.g. in government reports. The IPCC report is cited in this way as recommended in the report itself.
Reviewer 4 Report
The authors corrected most of the errors and addressed most of my comments.
However, the article could be a bit more refined, both in terms of content and style; figures should have better quality.
I also think that advanced statistical calculations are still lacking - the authors added an ANOVA analysis, but its results were not discussed in detail and there are no implications of these results.
Conclusions are still too vague and references have not been developed too much.
In my opinion, the manuscript still needs to be refined. After correcting and referring to the comments - it can be published.
Author Response
We have put the figures in the document anew, and they will also be uploaded separately. They are 2000x2000 png files, and should not be low quality anymore.
To address the concerns about statistics and uncertainty, we have:
- Gone over the whole paper and made it clearer that the discussion of differences between the databases throughout the text relate to the significant difference between them and added more elaboration.
- Added a supplementary table with confidence intervals for the emission factors and explanation of the calculation in the methods section.
- Added references in the text to the uncertainty seen in the confidence intervals, which supports the differences between the databases.
The conclusion has been rewritten.
We have added more references which provide more context, most of which are recent (mainly from 2022, and a few from 2023). But as originally commented: Some of the "older" references are key papers in this field, e.g. Poore and Nemecek (2018) and Willet et al. (2019) etc., and we do not believe references of this quality should be replaced because they are not the most recent.